# MoReL: Multi-omics Relational Learning

**Arman Hasanzadeh, Ehsan Hajiramezanali, Nick Duffield & Xiaoning Qian**
Department of Electrical and Computer Engineering, Texas A&M University
{armanihm,ehsanr,duffieldng,xqian}@tamu.edu

## Abstract

Multi-omics data analysis has the potential to discover hidden molecular inter-
actions, revealing potential regulatory and/or signal transduction pathways for
cellular processes of interest when studying life and disease systems. One of
critical challenges when dealing with real-world multi-omics data is that they may
manifest heterogeneous structures and data quality as often existing data may be
collected from different subjects under different conditions for each type of omics
data. We propose a novel deep Bayesian generative model to efficiently infer a
multi-partite graph that encodes molecular interactions across such heterogeneous
views, using a fused Gromov-Wasserstein (FGW) regularization between latent
representations of corresponding views for integrative analysis. With such an
optimal transport regularization in the deep Bayesian generative model, it not only
allows incorporating view-specific side information, either with graph-structured or
unstructured data in different views, but also increases the model flexibility with the
distribution-based regularization. This allows efficient alignment of heterogeneous
latent variable distributions to derive reliable interaction predictions compared to
the existing point-based graph embedding methods. Our experiments on several
real-world datasets demonstrate the enhanced performance of MoReL in inferring
meaningful interactions compared to existing baselines.

## 1 Introduction

Multi-view learning tries to fully leverage the information from multiple sources (i.e. different types of
omics data in molecular biology) and represents them in a shared embedding space, which is beneficial
for many downstream tasks with a limited number of training samples. In biomedical applications, the
shared embedding space also enables better understanding of the underlying biological mechanisms
by discovering interactions between different types of molecules, which is our focus in this paper.

Existing multi-omics data integration methods are limited in their applicability. First, most of them
attempt to derive low-dimensional embeddings of the input samples and are not designed to infer
a multi-partite graph that encodes the interactions across views. In unsupervised settings, matrix
factorization based methods, such as Bayesian Canonical Correlation Analysis (BCCA) (Klami et al.,
2013) and Multi-Omics Factor Analysis (MOFA) (Argelaguet et al., 2018), can achieve the similar
goal of cross-view relational learning but often through two-step procedures, in which the factor
loading parameters are used for downstream interaction analyses across views. Second, a very recent
relational inference for multi-view data integration, BayRel (Hajiramezanali et al., 2020), is built
on three strict assumptions, which may limit its practical application, including in multi-omics data
integration: 1) A graph of dependency between features of each view is available; 2) The input
dataset is complete on all views with no missing samples; 3) The samples in different views are
well-paired. While the first limitation might be solved by learning a graph using an *ad-hoc* technique,
the last two issues are common in many multi-omics data integration problems. Integrated samples
commonly have one or more views with various missing patterns. This is mostly due to limitations
of experimental designs or compositions from different data platforms. In addition, data might be
collected in different laboratories or the sample IDs are not available due to patient identification or
privacy/security concerns, leading to unpaired datasets. Apart from these, we might not have access
to *a priori* graph structured data in some view(s) as the nature of data might not be structured, or we
only have incomplete or very noisy prior knowledge. For such multi-omics data, leaving out such
a view may lose some complementary information while enforcing graph structures may lead to
degraded performances.

In this work, we propose a new **M**ulti-**o**mics **Re**lational **L**earning method, MoReL, based on the fused Gromov-Wasserstein (FGW) regularization, mitigating the dependency of multi-view learning on the aforementioned two assumptions. The proposed method contains four major contributions: 1) MoReL provides a new Bayesian multi-omics relational learning framework with efficient variational inference and is able to exploit non-linear transformations of data by leveraging deep learning models for either unstructured or graph-structured data; 2) MoReL learns a multi-partite graph across different features from multiple views using a FGW-based decoder, facilitating meaningful biological knowledge discovery from integrative multi-omics data analysis while accounting for arbitrarily permutation and/or transformation caused by processing features with different deep functions across the views; 3) MoReL can flexibly integrate both structured and unstructured heterogeneous views in one framework, in which only confident constraints need to be imposed to improve the model performance; 4) MoReL is able to integrate multiple views with unpaired samples and/or arbitrary sample-missing patterns.

## 2 RELATED WORKS

**Optimal transport**. There have been extensive efforts to utilize Gromov-Wasserstein (GW) discrepancy to solve the alignment problems in shape and object matching (Mémoli, 2009; 2011). A similar attempt has been made recently to investigate its potential for more diverse applications, such as aligning vocabulary sets between different languages (Alvarez-Melis & Jaakkola, 2018), and graph matching (Chowdhury & Mémoli, 2019; Vayer et al., 2018b; Xu et al., 2019b). Peyré et al. (2016) have proposed a fast Sinkhorn projection-based algorithm (Cuturi, 2013) to compute the entropy-regularized GW distance. Following this direction, Xu et al. (2019b) have replaced the entropy regularizer with a Bregman proximal term. To further reduce the computational complexity, the recursive GW distance (Xu et al., 2019a) and the sliced GW distance (Vayer et al., 2019) have been proposed. In Bunne et al. (2019), a pair of generative models are learned for incomparable spaces by defining an adversarial objective function based on the GW discrepancy. It imposes an orthogonal assumption on the transformation between the sample and its latent space. However, it can not incorporate the graph structured data. Similar to our model in this paper, Vayer et al. (2018a) and Xu et al. (2020) have proposed to impose the fused GW regularization in their objective functions by combining GW and Wasserstein discrepancies.

**Graph CCA (gCCA)**. To utilize *a priori* known information about geometry of the samples, gCCA methods (Chen et al., 2019; 2018) have been proposed to construct a dependency graph between *samples* and directly impose it into a regularizer. Similar to classical CCA, gCCA learns an unstructured shared latent representation. Unlike our MoReL, though, they can neither take advantage of the dependency graph between *features*, nor explicitly model relational dependency between *features* across views. Therefore, they rely on *ad-hoc* post-processing procedures to infer inter-relations.

**Graph representation learning**. Graph neural network architectures have been shown to be effective for link prediction (Hamilton et al., 2017; Kipf & Welling, 2016; Hasanzadeh et al., 2019; Hajiramezanali et al., 2019; Hasanzadeh et al., 2020) as well as matrix completion for recommender systems (Berg et al., 2017; Monti et al., 2017; Kalofolias et al., 2014; Ma et al., 2011). The first group of models is dealing with a single graph and is not able to deal with heterogeneous graphs, with multiple types of nodes and edges, and node attributes (Zhang et al., 2019). The second group utilizes the known item-item and user-user relationships and their attributes to complete the user-item rating matrix. However, they rely on two strict assumptions: 1) the inter-relation matrix is partially observed; and 2) both views have structured information. The proposed MoReL achieves robust multi-view learning without these assumptions, making it more practical in multi-omics data integration.

## 3 PRELIMINARIES

### 3.1 WASSERSTEIN DISTANCE

Wasserstein distance (WD) quantifies the geometric discrepancy between two probability distributions by measuring the minimal amount of "work" needed to move all the mass contained in one distribution onto the other (Solomon et al., 2015). More specifically, given two probability measures $\boldsymbol{\Lambda} \in \mathcal{P}(\mathbb{X})$ and $\boldsymbol{\Delta} \in \mathcal{P}(\mathbb{Y})$, and a transportation cost $c : \mathbb{X} \times \mathbb{Y} \to \mathbb{R}_+$, WD is the solution to the following

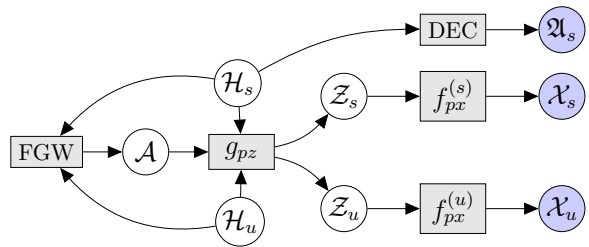

Figure 1: Graphical illustration of MoReL's generative flow with structured and unstructured views. DEC stand for decoder. The rest of variables and abbreviations are defined in the manuscript.

optimization problem:

$$\inf_{\pi \in \Pi(\mathbb{X} \times \mathbb{Y})} \mathbb{E}_{(\mathbf{x}, \mathbf{y}) \sim \pi}[c(\mathbf{x}, \mathbf{y})] = \inf_{\pi \in \Pi(\mathbb{X} \times \mathbb{Y})} \int c(\mathbf{x}, \mathbf{y}) \, d\pi(\mathbf{x}, \mathbf{y}),$$

where $\pi$ is the transport map and $\Pi(\mathbb{X} \times \mathbb{Y}) := \{\pi \in \mathcal{P}(\mathbb{X} \times \mathbb{Y}) \mid \int \pi(\mathbf{x}, \mathbf{y}) \, d\mathbf{y} = \mathbf{\Lambda}(\mathbf{x}), \int \pi(\mathbf{x}, \mathbf{y}) \, d\mathbf{x} = \mathbf{\Delta}(\mathbf{y})\}$ is the set of all admissible couplings. Assuming that the probability distributions are discrete, with probability mass functions $\sum_{i=1}^{n} a_i \delta_{\mathbf{x}_i}$ and $\sum_{j=1}^{m} b_j \delta_{\mathbf{y}_j}$, WD optimization could be simplified as follows:

$$\mathcal{D}_{\mathrm{W}}(\mathbf{\Lambda}, \mathbf{\Delta}) = \min_{\mathbf{T} \in \Pi(a, b)} \sum_{i=1}^{n} \sum_{j=1}^{m} T_{i,j} \, c(\mathbf{x}_i, \mathbf{y}_j),$$

where $T_{i,j}$ is an element of the transport matrix $\mathbf{T}$ whose row-wise and column-wise sums equal to $[a_i]_{i=1}^{n}$ and $[b_j]_{j=1}^{m}$, respectively.

### 3.2 GROMOV-WASSERSTEIN DISTANCE

Gromov-Wasserstein distance (GWD) has been proposed as a natural extension of WD when a meaningful transportation cost between the distributions cannot be defined. For example, when two distributions are defined in Euclidean spaces with different dimensions or more generally when $\mathbb{X}$ and $\mathbb{Y}$ are unaligned, i.e. when their features are not in correspondence (Vayer et al., 2019). Instead of measuring inter-domain distances, GWD measures the distance between pairs of samples in one domain and compares it to those in the other domain. More specifically, given two probability measures $\mathbf{\Lambda} \in \mathcal{P}(\mathbb{X})$ and $\mathbf{\Delta} \in \mathcal{P}(\mathbb{Y})$, as well as two domain-specific transportation costs $c^{(\mathbb{X})} : \mathbb{X} \times \mathbb{X} \to \mathbb{R}_+$ and $c^{(\mathbb{Y})} : \mathbb{Y} \times \mathbb{Y} \to \mathbb{R}_+$, GWD is the solution to the following optimization problem:

$$\inf_{\pi \in \Pi(\mathbb{X} \times \mathbb{Y})} \mathbb{E}_{(\mathbf{x}, \mathbf{y}) \sim \pi, (\mathbf{x}', \mathbf{y}') \sim \pi}[L(\mathbf{x}, \mathbf{x}', \mathbf{y}, \mathbf{y}')] = \inf_{\pi \in \Pi(\mathbb{X} \times \mathbb{Y})} \int \int L(\mathbf{x}, \mathbf{x}', \mathbf{y}, \mathbf{y}') \, d\pi(\mathbf{x}, \mathbf{y}) \, d\pi(\mathbf{x}', \mathbf{y}'),$$

where $L(\mathbf{x}, \mathbf{x}', \mathbf{y}, \mathbf{y}') = \| c^{(\mathbb{X})}(\mathbf{x}, \mathbf{x}') - c^{(\mathbb{Y})}(\mathbf{y}, \mathbf{y}') \|$, $\pi$ is the transport map, and $\Pi(\mathbb{X} \times \mathbb{Y}) := \{\pi \in \mathcal{P}(\mathbb{X} \times \mathbb{Y}) \mid \int \pi(\mathbf{x}, \mathbf{y}) \, d\mathbf{y} = \mathbf{\Lambda}(\mathbf{x}), \int \pi(\mathbf{x}, \mathbf{y}) \, d\mathbf{x} = \mathbf{\Delta}(\mathbf{y})\}$ is the set of all admissible couplings. Likewise, this can be derived for discrete distributions with probability mass functions $\sum_{i=1}^{n} a_i \delta_{\mathbf{x}_i}$ and $\sum_{j=1}^{m} b_j \delta_{\mathbf{y}_j}$, as follows:

$$\mathcal{D}_{\mathrm{GW}}(\mathbf{\Lambda}, \mathbf{\Delta}) = \min_{\mathbf{T} \in \Pi(a, b)} \sum_{i,i'=1}^{n} \sum_{j,j'=1}^{m} T_{i,j} \, T_{i',j'} \, L(\mathbf{x}_i, \mathbf{x}_{i'}, \mathbf{y}_j, \mathbf{y}_{j'}), \tag{1}$$

where $T_{i,j}$ is an element of transport matrix $\mathbf{T}$ whose row-wise and column-wise sums equal to $[a_i]_{i=1}^{n}$ and $[b_j]_{j=1}^{m}$, respectively.

## 4 METHOD

### 4.1 PROBLEM FORMULATION AND NOTATIONS

We propose a novel hierarchical generative model for multi-omics data integration that incorporates view-specific structure information when it is available. Given observations from structured and

unstructured views, our model, Multi-omics Relational Learning (MoReL), aims to infer the *inter-relations* among entities, i.e. features, across all of the views. More specifically, assume that multiple views, $\mathcal{V}$, of data are given. Without loss of generality, we assume that the structure information, provided as a graph, is available for some of the views $\mathcal{V}_s \subset \mathcal{V}$, and the remaining views $\mathcal{V}_u = \mathcal{V} \setminus \mathcal{V}_s$ are unstructured. We note that every structure could be represented as a graph. For example, image and sequential data could be represented over grid and directed path graphs, respectively.

We represent the set of graphs for structured views by $\mathfrak{G}_s = \{\mathcal{G}^{(v)}\}_{v \in \mathcal{V}_s}$ and their adjacency matrices by $\mathfrak{A}_s = \{\mathbf{A}^{(v)}\}_{v \in \mathcal{V}_s}$. We also define $\mathcal{X}_s = \{\mathbf{X}^{(v)}\}_{v \in \mathcal{V}_s}$ as the set of node attributes for structured views, and $\mathcal{X}_u = \{\mathbf{X}^{(v)}\}_{v \in \mathcal{V}_u}$ as the set of data for unstructured views. Moreover, $N_v$ denotes the number of nodes in structured views and number of features for unstructured views. MoReL infers the interactions among the nodes in $\mathfrak{G}_s$ and features in $\mathcal{X}_u$. We represent these inter-relations by a multi-partite graph with $\sum_{v \in \mathcal{V}} N_v$ nodes and a multi-adjacency tensor $\mathcal{A} = \{\mathbf{A}^{(vv')}\}_{v, v' \in \mathcal{V}, v \neq v'}$, where $\mathbf{A}^{(vv')}$ is the $N_v \times N_{v'}$ bi-adjacency matrix between views $v$ and $v'$.

## 4.2 MoReL GENERATIVE MODEL

We define a hierarchical Bayesian model for MoReL with three sets of latent variables: 1) $\mathcal{H} = \mathcal{H}_s \cup \mathcal{H}_u = \{\mathbf{H}^{(v)}\}_{v \in \mathcal{V}_s \cup \mathcal{V}_u}$, which captures the (hidden) structural information; 2) $\mathcal{A}$, which encodes the interaction among features across views; and 3) $\mathcal{Z} = \mathcal{Z}_s \cup \mathcal{Z}_u = \{\mathbf{Z}^{(v)}\}_{v \in \mathcal{V}_s \cup \mathcal{V}_u}$, which summarizes the feature/attribute specific information. The joint probability of observations and latent variables factorizes as follows:

$$
\begin{aligned}
p_\theta(\mathcal{X}_u, \, &\mathcal{X}_s, \, \mathfrak{A}_s, \, \mathcal{H}, \, \mathcal{A}, \, \mathcal{Z}) = \\
&p_{\theta_x}(\mathcal{X}_u \,|\, \mathcal{Z}_u) \, p_{\theta_x}(\mathcal{X}_s \,|\, \mathcal{Z}_s) \, p_{\theta_g}(\mathfrak{A}_s \,|\, \mathcal{H}_s) \, p_{\theta_z}(\mathcal{Z} \,|\, \mathcal{H}, \, \mathcal{A}) \, p_{\theta_a}(\mathcal{A} \,|\, \mathcal{H}) \, p(\mathcal{H}).
\end{aligned}
\tag{2}
$$

Figure 1 depicts the generative model of MoReL with structured and unstructured views. In the following subsections, we define different parts of the generative and inference model.

### 4.2.1 OPTIMAL TRANSPORT FOR MULTI-PARTITE GRAPH DECODER

In this subsection, we define the generative distribution of the multi-adjacency tensor, $\mathcal{A}$. We note that inferring $\mathcal{A}$ is the main goal of our model. Given the structural latent variables $\mathcal{H}$, we introduce a fused Gromov-Wasserstein (FGW) distance based mapping to generate $\mathcal{A}$. FGW refers to distance metrics defined by combining WD and GWD, which has been proposed to compare structured distributions (Vayer et al., 2018b; Chen et al., 2020). Considering graphs with node attributes as structured distributions, WD compares node distributions in two graphs (i.e, node similarity), GWD measures the distance between pairs of nodes in one graph and compares it to those in the other (i.e., edge/path similarity).

**FGW distance.** Given two structured probability distributions, $\mathbf{\Lambda} \in \mathcal{P}(\mathbb{X})$ and $\mathbf{\Delta} \in \mathcal{P}(\mathbb{Y})$, FGW is defined as follows:

$$
\begin{aligned}
\mathcal{D}_{\mathrm{FGW}}(\mathbf{\Lambda}, \, \mathbf{\Delta}) &= \alpha \, \mathcal{D}_{\mathrm{W}}(\mathbf{\Lambda}, \, \mathbf{\Delta}) + \beta \, \mathcal{D}_{\mathrm{GW}}(\mathbf{\Lambda}, \, \mathbf{\Delta}) \\
&= \alpha \inf_{\pi_w \in \Pi(\mathbb{X} \times \mathbb{Y})} \mathbb{E}_{(\mathbf{x}, \mathbf{y}) \sim \pi_w}[c^{(\mathbb{X}\mathbb{Y})}(\mathbf{x}, \, \mathbf{y})] \\
&\quad + \beta \inf_{\pi_{gw} \in \Pi(\mathbb{X} \times \mathbb{Y})} \mathbb{E}_{(\mathbf{x}, \mathbf{y}), (\mathbf{x}', \mathbf{y}') \sim \pi_{gw}}[\|\, c^{(\mathbb{X})}(\mathbf{x}, \, \mathbf{x}') - c^{(\mathbb{Y})}(\mathbf{y}, \, \mathbf{y}') \,\|],
\end{aligned}
\tag{3}
$$

where $\alpha, \beta \in [0, 1]$ are scalar hyper-parameters, $\Pi(\mathbb{X} \times \mathbb{Y})$ is the set of all admissible couplings between $\mathbf{\Lambda}$ and $\mathbf{\Delta}$, and $c^{(\mathbb{X}\mathbb{Y})}, c^{(\mathbb{X})}$, and $c^{(\mathbb{Y})}$ are corresponding transportation cost functions. $\mathcal{D}_{\mathrm{FGW}}$ can be further simplified by choosing $\pi_w$ to be equal to $\pi_{gw}$ (Chen et al., 2020).

**Relational learning via FGW.** We are interested in aligning the nodes/features in every pair of views, i.e. $(v, v')$. Hence, we will have a FGW distance based decoder for every pair of views, in which each view independently belongs to either structured or unstructured views, i.e. $(v, v') \in \mathcal{V}$. To that end, we first define the transportation cost functions $c^{(vv')}$ and $c^{(v)}$, and then approximate $\mathcal{D}_{\mathrm{FGW}}$. We define the (inter-)cost function for the first term of FGW, i.e. $\mathcal{D}_{\mathrm{W}}$, as follows:

$$
c^{(vv')}(\mathbf{H}_{i,:}^{(v)}, \, \mathbf{H}_{j,:}^{(v')}) = 1 - \sigma\left(\mathbf{H}_{i,:}^{(v)} (\mathbf{H}_{j,:}^{(v')})^T\right); \qquad v, v' \in \mathcal{V}, \tag{4}
$$

where $\sigma$ denotes the sigmoid function, and $\mathbf{H}_{i,:}^{(v)}$ represents the structural latent variable of node/feature $i$ in view $v$. To calculate the $\mathcal{D}_{\text{GW}}$, we define two different transportation costs based on the nature of the inputs. For the structured views, we define the cost function as a combination of the shortest path distance from graph and the distance between structural latent variables. More specifically, given the normalized shortest path distance matrix between every pair of nodes in the input graph $\mathbf{D}^{(v)}$:

$$c^{(v)}(\mathbf{H}_{i,:}^{(v)}, \mathbf{H}_{j,:}^{(v)}) = \mathbf{D}^{(v)} \odot \left(1 - \sigma\left(\mathbf{H}_{i,:}^{(v)}(\mathbf{H}_{j,:}^{(v)})^T\right)\right); \qquad \text{for } v \in \mathcal{V}_s,$$

where $\odot$ denotes the Hadamard product. This construction ensures both graph and attributes information are incorporated in the distance function. For unstructured views, we define the cost function between two features as follows:

$$c^{(v)}(\mathbf{H}_{i,:}^{(v)}, \mathbf{H}_{j,:}^{(v)}) = 1 - \sigma\left(\mathbf{H}_{i,:}^{(v)}(\mathbf{H}_{j,:}^{(v)})^T\right); \qquad \text{for } v \in \mathcal{V}_u.$$

Noting the definitions of WD and GWD in Section 3, we rewrite $\mathcal{D}_{\text{FGW}}$ between two views of data with shared transport matrixas follows:

$$\mathcal{D}_{\text{FGW}}\left(p(\mathbf{H}^{(v)}), p(\mathbf{H}^{(v')})\right) =$$
$$\sum_{i=1}^{N_v}\sum_{j=1}^{N_{v'}} \min_{\mathbf{T}_{gw}^{(vv')} \in \Pi} \sum_{\mathbf{H}_{i,:}^{(v)}, \mathbf{H}_{j,:}^{(v')}, \mathbf{H}_{i,:}^{(v)'}, \mathbf{H}_{j,:}^{(v')'}} \left[\alpha\, c^{(vv')}(\mathbf{H}_{i,:}^{(v)}, \mathbf{H}_{j,:}^{(v')}) + \right. \tag{5}$$
$$\left. \beta \parallel c^{(v)}(\mathbf{H}_{i,:}^{(v)}, \mathbf{H}_{i,:}^{(v)'}) - c^{(v')}(\mathbf{H}_{j,:}^{(v')}, \mathbf{H}_{j,:}^{(v')'}) \parallel \right].$$

To approximate the FGW distance, we first deploy GW algorithm in equation (1) to obtain $\mathbf{T}_{gw}^{(vv')}$ and $\mathcal{D}_{\text{GW}}$, and then utilize $\mathbf{T}_{gw}^{(vv')}$ along with the defined transportation cost $c^{(vv')}$ to calculate Wasserstein distance term in $\mathcal{D}_{\text{FGW}}$ (Chen et al., 2020). The pseudo-code in Algorithm 1 (Appendix A.1) provides the details of the FGW distance calculation procedure. Please note that we use the same Sinkhorn solver as in Chen et al. (2020) and Alvarez-Melis & Jaakkola (2018).

We further can generate $\mathcal{A}$ for every pair of views based on $\mathbf{T}_{gw}^{(vv')}$ as follows:

$$p(\mathcal{A} \,|\, \mathcal{H}) = \prod_{\substack{v,v' \in \mathcal{V} \\ v \neq v'}} p(\mathbf{A}^{(vv')} \,|\, \mathbf{H}^{(v)}, \mathbf{H}^{(v')}) = \prod_{\substack{v,v' \in \mathcal{V} \\ v \neq v'}} \text{Ber}\left(\mathbf{A}^{(vv')} \,|\, \gamma\, \mathbf{T}_{gw}^{(vv')}/\max(\mathbf{T}_{gw}^{(vv')})\right), \quad (6)$$

where $\gamma \in [0,1]$ is a normalizing hyper-parameter, and $\text{Ber}$ is short for Bernoulli. We note that the sum of the elements in each of the transport matrices $\mathbf{T}_{gw}^{(vv')}$ equals to one. Hence each of its elements has a small value. Therefore, we normalize the transport matrices (as $\gamma\, \mathbf{T}_{gw}^{(vv')}/\max(\mathbf{T}_{gw}^{(vv')})$) to avoid very sparse and trivial solutions. To use the reparametrization trick during training, we sample from concrete relaxation of Bernoulli (Gal et al., 2017). We emphasize that our proposed FGW-based decoder is the key in aligning features/nodes across structured and unstructured views via accurate and efficient *distribution matching* scheme.

### 4.2.2 Prior construction and likelihoods

**Prior**. We impose independent zero-mean unit-variance Gaussian priors on elements of $\mathcal{H}$. The prior for $\mathcal{Z}$ is a multivariate Gaussian distribution whose mean and diagonal covariance matrix are constructed from the inferred multi-partite graph and the structural latent variable $\mathcal{H}$. We use two graph neural networks (GNNs) $g_{pz}^{(\mu)}$ and $g_{pz}^{(\sigma)}$ to map $\mathcal{H}$ and $\mathcal{A}$ to the parameters of $p_{\theta_z}(\mathcal{Z})$. Specifically,

$$p_{\theta_z}(\mathcal{Z} \,|\, \mathcal{H}, \mathcal{A}) = \prod_{v \in \mathcal{V}_s} \prod_{i=1}^{N_v} p_{\theta_z}(\mathbf{Z}_{i,:}^{(v)} \,|\, \mathcal{H}, \mathcal{A}); \qquad p_{\theta_z}(\mathbf{Z}_{i,:}^{(v)} \,|\, \mathcal{H}, \mathcal{A}) = \mathcal{N}(\boldsymbol{\mu}_{pz}^{(v,i)}, \boldsymbol{\sigma}_{pz}^{(v,i)}),$$
$$\text{with} \qquad [\boldsymbol{\mu}_{pz}^{(v,i)}]_{v,i} = g_{pz}^{(\mu)}(\mathcal{H}, \mathcal{A}), \qquad [\boldsymbol{\sigma}_{pz}^{(v,i)}]_{v,i} = g_{pz}^{(\sigma)}(\mathcal{H}, \mathcal{A}).$$

We note that in this setting, $\mathcal{H}$ is considered as node attributes of the multi-partite interaction graph.

**Likelihood of observations**. To reconstruct the input graphs in the structured views, we assume that the views and edges are conditionally independent. More specifically, we employ an inner-product decoder as follows:

$$p_{\theta_g}(\mathfrak{A}_s \,|\, \mathcal{H}_s) = \prod_{v \in \mathcal{V}_s} \prod_{i,j=1}^{N_v} p_{\theta_g}\left(\mathbf{A}_{i,j}^{(v)} \,|\, \mathbf{H}_{i,:}^{(v)}, \, \mathbf{H}_{j,:}^{(v)}\right);$$

$$p_{\theta_g}\left(\mathbf{A}_{i,j}^{(v)} \,|\, \mathbf{H}_{i,:}^{(v)}, \mathbf{H}_{j,:}^{(v)}\right) = \mathrm{Ber}\left(\sigma(\mathbf{H}_{i,:}^{(v)} \, (\mathbf{H}_{j,:}^{(v)})^T)\right).$$

To generate the features in unstructured views and node attributes in structured views, we assume that views are conditionally independent. Hence we can expand the feature reconstruction terms in the the equation (2) as follows:

$$p_{\theta_x}(\mathcal{X}_u \,|\, \mathcal{Z}_u) = \prod_{v \in \mathcal{V}_u} p_{\theta_x}(\mathbf{X}^{(v)} \,|\, \mathbf{Z}^{(v)}), \qquad p_{\theta_x}(\mathcal{X}_s \,|\, \mathcal{Z}_s) = \prod_{v \in \mathcal{V}_s} p_{\theta_x}(\mathbf{X}^{(v)} \,|\, \mathbf{Z}^{(v)}).$$

We note that $p_{\theta_x}$ could also be view specific depending on whether the node attributes/features in a view are discrete or continuous. In our experiments, we have deployed the Gaussian likelihood with the unit variance. The mapping from $\mathcal{Z}$ to the parameters of $p_{\theta_x}(\mathcal{X})$, in our case, the mean of the Gaussian distribution, can be any highly expressive function such as neural networks. We denote these functions by $f_{px}^{(v,s)}$ and $f_{px}^{(v,u)}$.

### 4.3 INFERENCE NETWORK AND LEARNING

**Posterior**. We model the posterior of the structural latent variables as a Gaussian distribution and infer its parameters independently for each view. More specifically,

$$q_{\phi_h}(\mathcal{H}_u \,|\, \mathcal{X}_u) = \prod_{v \in \mathcal{V}_u} q_{\phi_h}(\mathbf{H}^{(v)} \,|\, \mathbf{X}^{(v)}), \qquad q_{\phi_h}(\mathcal{H}_s \,|\, \mathcal{X}_u, \mathfrak{A}_s) = \prod_{v \in \mathcal{V}_s} q_{\phi_h}(\mathbf{H}^{(v)} \,|\, \mathbf{X}^{(v)}, \mathbf{A}^{(v)}).$$

We use two GNNs for each structured view, $\{g_{qh}^{(\mu,v)}(\mathbf{X}^{(v)}, \mathbf{A}^{(v)}), g_{qh}^{(\sigma,v)}(\mathbf{X}^{(v)}, \mathbf{A}^{(v)})\}_{v \in \mathcal{V}_s}$, and two fully connected neural networks per unstructured view, $\{f_{qh}^{(\mu,v)}(\mathbf{X}^{(v)}), f_{qh}^{(\sigma,v)}(\mathbf{X}^{(v)})\}_{v \in \mathcal{V}_u}$, to map inputs to the mean and variance of the posteriors. We consider the variational distribution of $\mathcal{Z}$ to be a multivariate Gaussian distribution, and it is factorized as follows:

$$q_{\phi_z}(\mathcal{Z}_u \,|\, \mathcal{X}_u) = \prod_{v \in \mathcal{V}_u} q_{\phi_z}(\mathbf{Z}^{(v)} \,|\, \mathbf{X}^{(v)}), \qquad q_{\phi_z}(\mathcal{Z}_s \,|\, \mathcal{X}_u, \mathfrak{A}_s) = \prod_{v \in \mathcal{V}_s} q_{\phi_z}(\mathbf{Z}^{(v)} \,|\, \mathbf{X}^{(v)}, \mathbf{A}^{(v)}).$$

We use two GNNs per structured view, $\{g_{qz}^{(\mu,v)}(\mathbf{X}^{(v)}, \mathbf{A}^{(v)}), g_{qz}^{(\sigma,v)}(\mathbf{X}^{(v)}, \mathbf{A}^{(v)})\}_{v \in \mathcal{V}_s}$, and two fully connected neural networks for each unstructured view, $\{f_{qz}^{(\mu,v)}(\mathbf{X}^{(v)}), f_{qz}^{(\sigma,v)}(\mathbf{X}^{(v)})\}_{v \in \mathcal{V}_u}$, in the same fashion as $q_{\phi_h}$ to infer parameters of $q_{\phi_z}$.

**Objective function**. Having defined the prior and posterior distributions as well as the likelihood, we write the overall loss function as the sum of the negative variational ELBO and FGW regularization terms. Specifically,

$$\begin{aligned}
\mathcal{L} &= -\,\mathrm{ELBO} + \mathcal{L}_{\mathrm{FGW}} \\
&= \mathbb{E}_{q_{\phi_z}(\mathcal{Z}_u, \mathcal{H}_u \,|\, \mathcal{X}_u)} \log p_\theta(\mathcal{Z}_u \,|\, \mathcal{A}, \mathcal{H}) + \mathbb{E}_{q_{\phi_z}(\mathcal{Z}_s, \mathcal{H}_s \,|\, \mathcal{X}_s, \mathfrak{A}_s)} \log p_\theta(\mathcal{Z}_s \,|\, \mathcal{A}, \mathcal{H}) \\
&\quad - \mathbb{E}_{q_{\phi_z}(\mathcal{Z}_u \,|\, \mathcal{X}_u)} \log q_{\phi_z}(\mathcal{Z}_u \,|\, \mathcal{X}_u) - \mathbb{E}_{q_{\phi_z}(\mathcal{Z}_s \,|\, \mathcal{X}_s, \mathfrak{A}_s)} \log q_{\phi_z}(\mathcal{Z}_s \,|\, \mathcal{X}_s, \mathfrak{A}_s) \\
&\quad + \mathbb{E}_{q_{\phi_h}(\mathcal{H}_u \,|\, \mathcal{X}_u)} \log p(\mathcal{H}_u) + \mathbb{E}_{q_{\phi_h}(\mathcal{H}_s \,|\, \mathcal{X}_s, \mathfrak{A}_s)} \log p(\mathcal{H}_s) \\
&\quad - \mathbb{E}_{q_{\phi_h}(\mathcal{H}_u \,|\, \mathcal{X}_u)} \log q_{\phi_h}(\mathcal{H}_u \,|\, \mathcal{X}_u) - \mathbb{E}_{q_{\phi_h}(\mathcal{H}_s \,|\, \mathcal{X}_s, \mathfrak{A}_s)} \log q_{\phi_h}(\mathcal{H}_s \,|\, \mathcal{X}_u, \mathfrak{A}_s)) \qquad (7) \\
&\quad + \mathbb{E}_{q_{\phi_z}(\mathcal{Z}_u \,|\, \mathcal{X}_u)} \log p_{\theta_x}(\mathcal{X}_u \,|\, \mathcal{Z}_u) + \mathbb{E}_{q_{\phi_z}(\mathcal{Z}_s \,|\, \mathcal{X}_s, \mathfrak{A}_s)} \log p_{\theta_x}(\mathcal{X}_s \,|\, \mathcal{Z}_s) \\
&\quad + \mathbb{E}_{q_{\phi_h}(\mathcal{H}_s \,|\, \mathcal{X}_s, \mathfrak{A}_s)} \log p_{\theta_g}(\mathfrak{A}_s \,|\, \mathcal{H}_s) + \sum_{v \in \mathcal{V}} \sum_{\substack{v' \in \mathcal{V} \\ v' \neq v}} \mathcal{D}_{\mathrm{FGW}}\left(p(\mathbf{H}^{(v)}), \, p(\mathbf{H}^{(v')})\right).
\end{aligned}$$

While, as mentioned previously, we use the Sinkhorn algorithm to calculate the $\mathcal{D}_{\mathrm{FGW}}$, the overall loss is optimized using stochastic gradient descent based optimization algorithms such as Adam (Kingma & Ba, 2014).

## 5 EXPERIMENTS

### 5.1 DATASETS AND EVALUATION METRICS

**Datasets**. We use the same datasets as BayReL (Hajiramezanali et al., 2020), i.e. microbiome-metabolite interactions in cystic fibrosis (CF) and gene-drug interactions in precision medicine. Dataset description and graph construction procedure are detailed in Appendix A.2. We want to emphasize that although these datasets have structured views, have no missing samples and their samples are completely paired, in many real-world cases these assumptions are not satisfied. These datasets were chosen merely to get a better understanding of the advantages of MoReL specially compared to BayReL. We evaluate MoReL in different settings. More specifically, we demonstrate the performance of MoReL when: 1) one or both views are unstructured, 2) there are missing samples, and 3) samples are not paired. Furthermore, we have a comprehensive comparison with BayReL when both views are structured.

**Evaluation metrics**. To quantify the performance of the methods, we use the same evaluation metrics as the ones introduced in BayReL. Since in these datasets, the true negatives, i.e. non-interactions, are not known; and there are only a small subset of true positives, i.e. true interactions, well-known classification metrics cannot be used for evaluation. Therefore, positive accuracy and negative accuracy have been defined to evaluate microbiome-metabolite experiments. Positive accuracy refers to the accuracy of identifying validated interactions with *P. aeruginosa*. Negative accuracy exploits the fact that there should not be any common metabolite targets between known anaerobic microbes (*Veillonella*, *Fusobacterium*, *Prevotella*, and *Streptococcus*) and notable pathogen *P. aeruginosa*. Let $\mathcal{B}$ denote the set of all microbes and $\mathcal{A}_1$ and $\mathcal{A}_2$ represent two disjoint sets of metabolites. Negative accuracy is defined as $1 - \frac{\sum_{i \in \mathcal{A}_1} \sum_{j \in \mathcal{A}_2} \sum_{l \in \mathcal{B}} \mathbb{1}(i \text{ and } j \text{ are connected to } l)}{|\mathcal{A}_1| \times |\mathcal{A}_2| \times |\mathcal{B}|}$, where $\mathbb{1}(\cdot)$ is the indicator function. Having both higher positive and negative accuracy is desired.

For precision medicine, we compare the prediction sensitivity of identifying known interactions in the test sets while tracking the average density of the overall constructed graphs. We note that inferring very dense graphs would lead to high prediction sensitivity as it will includes most of the possible interactions. Therefore, tracking the sparsity of the inferred graphs is the key to properly evaluate the models' capability in predicting meaningful interactions.

### 5.2 BASELINES AND EXPERIMENTAL SETUPS

**Baselines**. We compare MoReL with three baselines including Spearman's Rank Correlation Analysis (**SRCA**), **BCCA** (Klami et al., 2013), and **BayReL**. While SRCA applies to raw data, BCCA first finds low-dimensional latent representations of views via matrix factorization and then the interactions are discovered based on the correlation between representations. BCCA and SRCA could not incorporate the structure of data and need a two-step procedure to infer the interaction between features across the views. In contrast, BayReL is able to use the structure of data and infer the relations without any *ad-hoc* post-processing procedure. However, BayReL suffers from three strict assumptions: 1) All views of data are structured; 2) There are no missing samples in any views; and 3) Samples are paired, i.e. the ID of samples are known. We emphasize that MoReL is the very first model that not only can infer interactions across structured and unstructured views but also is able to handle missing and unpaired samples in different domains, making it more applicable in real-world multi-omics data integration. A widely used method for multi-omics data integration is MOFA (Argelaguet et al., 2018). The mathematical modeling of MOFA is the same as BCCA except for the data likelihood part. While BCCA only supports continuous data, MOFA can have discrete likelihoods. Since our datasets do not have discrete features, we are only reporting BCCA results.

**Hyper-parameters**. In all of our experiments, to have a fair comparison, architectural hyper-parameters (i.e. number of layers and number of neurons) were set to be the same as in BayReL. Other hyper-parameters that are unique to MoReL were tuned using the validation set. More specifically, the number of hidden layers as well as their dimensions are the same for the corresponding functions in both structured and unstructured views. We use graph convolutional layers (Kipf & Welling, 2017) for structured views and fully connected layers for unstructured views except for reconstructing $\mathcal{X}$ from $\mathcal{Z}$, for which we use fully connected layers in all of the views. The mapping from inputs to the mean and variance parameters of $\mathcal{H}$ are two 2-layer neural networks (16 and 8 dimensional layers)

Table 1: Comparison of positive accuracy (in %) on CF dataset at negative accuracy of $> 97\%$.

|  | SRCA | BCCA | MoReL $_{uu}$ | MoReL $_{us}$ |
|---|---|---|---|---|
| Positive accuracy | 26.41 | $28.30 \pm 3.21$ | $56.16 \pm 1.85$ | $63.77 \pm 1.11$ |

with a shared first layer for each view. We use two 2-layer neural networks (16 and 8 dimensional layers) with a shared first layer for each view for the mapping from $\mathcal{H}$ to the mean and variance of $\mathcal{Z}$. We use a 3-layer fully connected neural network (8 and 16 dimensional hidden layers) for each view as the reconstruction function mapping $\mathcal{Z}$ to $\mathcal{X}$. The temperature for relaxed Bernoulli distribution is set to 0.3. The normalizing parameter $\gamma$ in equation 6 is 0.9 while $\alpha$ and $\beta$ in $\mathcal{D}_{\text{FGW}}$ are set to 1 and 0.5, respectively. We used the exponential decaying learning rate with the decay rate of 0.01 and initial learning rate of 0.01. All of our results are averaged over multiple runs with different random seeds. We have implemented MoReL and all the competing methods in Tensorflow (Abadi et al., 2015). All the experiments are performed on a workstation with a single NVIDIA P100 GPU.

## 5.3 DISCUSSION, DATASETS WITH UNSTRUCTURED VIEWS

Table 1 shows the performance of three variants of MoReL and competing methods for microbiome-metabolite data integration with the CF data. In these experiments, we assume that the samples are paired and all are available in both views. In MoReL $_{uu}$, we report the results when both views are unstructured. In MoReL $_{us}$, we have the graph of interactions between microbiomes while the metabolite view is assumed to be unstructured. Comparing MoReL $_{uu}$ and baselines that do not incorporate any graph-structured data as input, we observe an almost 30% improvement in positive accuracy while maintaining higher than 97% negative accuracy. This demonstrates that our proposed MoReL, even without any structural information, is effective in inferring meaningful interactions. Further incorporating the network between microbiomes (i.e. MoReL $_{us}$) leads to a 37% and 7% improvement compared to the baselines and MoReL $_{uu}$, respectively. This shows not only the importance of incorporating view-specific side information, but also the effectiveness of FGW-based decoder in aligning structured and unstructured views. Further results on interpretability and robustness of MoReL on CF dataset is provided in Appendix A.3.

The results for prediction sensitivity of two variants of MoReL and competing methods in the precision medicine experiments are shown in Table 2. We observe that both MoReL $_{uu}$, where both views are unstructured, and MoReL $_{us}$, where the graph structure between genes is given, consistently outperform the baselines by a significant margin in graphs with different densities. This proves that MoReL is able to learn meaningful relations both in sparse and dense graphs. Comparing the results for MoReL $_{us}$ and BCCA, the difference between their performance increases as the the density of the bipartite graph increases, showing that MoReL $_{us}$ can identify gene-drug interactions more robustly.

## 5.4 COMPARISON WITH BAYREL

While the primary goal of experiments so far was showing the effectiveness of MoReL in integrating unstructured and structured views, here we investigate the advantages of MoReL over BayReL.

**All structured**. As mentioned earlier in the manuscript, BayReL assumes that all of the views are structured. To show the expressive power of MoReL, we train it in the same setting as BayReL where all of the views are structured. Particularly, we assume that for CF dataset both metabolites network and microbiome network are observed in the microbiome-metabolite experiment. Also, in precision medicine experiment both drug network and gene regulatory network are known *a priori*. For fair

Table 2: Comparison of prediction sensitivity (in %) in the precision medicine experiment.

| Avg. degree | 0.10 | 0.15 | 0.20 | 0.25 | 0.30 | 0.40 | 0.50 |
|---|---|---|---|---|---|---|---|
| SRCA | 8.03 | 12.00 | 17.15 | 20.70 | 26.85 | 34.93 | 45.79 |
| BCCA | $9.65 \pm 0.75$ | $14.34 \pm 0.06$ | $18.96 \pm 0.42$ | $23.29 \pm 0.52$ | $28.22 \pm 0.66$ | $38.02 \pm 2.15$ | $46.88 \pm 1.88$ |
| MoReL $_{uu}$ | $11.29 \pm 0.16$ | $15.74 \pm 0.62$ | $21.21 \pm 0.81$ | $26.20 \pm 1.10$ | $30.47 \pm 1.07$ | $39.05 \pm 0.75$ | $50.19 \pm 0.19$ |
| MoReL $_{us}$ | $12.79 \pm 0.39$ | $17.51 \pm 2.21$ | $22.82 \pm 1.01$ | $29.58 \pm 1.08$ | $35.05 \pm 1.27$ | $45.74 \pm 1.75$ | $53.16 \pm 0.96$ |

comparison, we set the number of layers as well as hidden dimensions to be the same in both models. We train MoReL with the exponential decaying learning rate with the initial rate of $0.01$ and decay rate of $0.001$ for 120 training epochs. For BayReL, we use the setting reported in Hajiramezanali et al. (2020). The results for CF and precision medicine are summarized in Tables 3 and 4. We see that MoReL outperforms BayReL on CF dataset by a margin of $7\%$ which indicates that knowing the metabolic pathways can greatly improve interaction learning. In precision medicine experiment, we observe a consistent $2\%$ improvement by MoReL compared to BayReL.

We emphasize that the declined performance of $\text{MoReL}_{uu}$ and $\text{MoReL}_{us}$ (shown in Tables 1 and 2) compared to BayReL is expected, as they uses less information than BayReL. Incorporating this extra information in MoReL enhances its performance substantially. Note that BayReL is bound to use the same set of functions for all views to account for arbitrarily rotations and transformations, which limits its expressive power. However, the FGW based decoder in MoReL allows to have different processing functions for each view. We argue that this increases the expressive power and plays the key role in enhancing the performance.

Table 3: Positive accuracy (%) on CF dataset.

|  | BayReL | $\text{MoReL}_{ss}$ |
|---|---|---|
| Positive Acc. | $82.70 \pm 4.70$ | $89.50 \pm 3.29$ |

Table 4: Prediction sensitivity (%) in the precision medicine experiment.

| Avg. degree | BayReL | $\text{MoReL}_{ss}$ |
|---|---|---|
| 0.4 | $47.90 \pm 0.43$ | $49.24 \pm 1.64$ |
| 0.5 | $56.76 \pm 0.50$ | $58.92 \pm 0.40$ |

**Paired vs. unpaired**. To show that MoReL can handle unpaired input samples, we perform an ablation study on CF dataset. We reverse the order of samples in metabolite view while keeping the order of samples in microbiome. We report the performance of BayReL and $\text{MoReL}_{us}$ where we don't use the structure of metabolite view. The results are shown in Table 5. While MoReL performed virtually the same as a completely paired scenario (shown in Table 1), BayReL's performance drastically declined. We note that the reported negative accuracy is the best one achieved by BayReL.

**Missing samples**. We should again point out that in a setting where all views are structured but the number of node attributes are not the same in different views, BayReL cannot be deployed (as it uses the same processing functions for all views). To see how $\text{MoReL}_{us}$ performs in such a scenario, we randomly remove $10\%$ of samples in metabolite view of CF dataset. MoReL achieves positive accuracy

Table 5: Positive accuracy (%) on CF dataset with unpaired samples.

|  | BayReL | $\text{MoReL}_{us}$ |
|---|---|---|
| Positive Acc. | 31.56 | $63.24 \pm 2.13$ |
| Negative Acc. | 72 | 97 |

(in %) of $61.36 \pm 3.74$ with negative accuracy of $97\%$. This again shows the robustness of FGW-based decoder in aligning nodes with different number of samples.

**Computational complexity**. We have also benchmarked computational complexity of MoReL and BayReL by tracking their runtime on CF dataset on the same hardware. While BayReL takes $0.6$ seconds per training epoch, MoReL takes $2.7$ seconds per training epoch. This is due to the computational overhead caused by deploying FGW-based decoder. Considering the model flexibility and significant prediction performance improvement, such computational overhead is acceptable.

## 6 CONCLUSIONS

We have proposed MoReL, a novel Bayesian deep generative model that efficiently infers hidden molecular relations across heterogeneous views of data. By using a fused Gromov-Wasserstein based decoder, MoReL addresses several main shortcomings of the state-of-the-art omics data integration model. Specifically, MoReL can: 1) integrate both structured and unstructured omics datasets while accounting for arbitrarily permutation and/or transformation caused by processing features with different deep functions across the views; 2) handle unpaired samples across the views of data; 3) combine multiple views from different data sources with any number of missing samples. Our experiments on two real-world datasets have demonstrated substantial improvement in inferring meaningful relations as well as improving prediction sensitivity compared to the competing methods. MoReL has shown the promising potential for multi-view learning, in particular multi-omics data integration for biological knowledge discovery, when facing heterogeneous data from different views.

ACKNOWLEDGMENTS

The presented materials are based upon the work supported in part by the National Science Foundation under Grants CCF-1553281, CCF-1934904, DMR-2119103, ECCS-1839816, IIS-1812641, IIS-1848596, and OAC-1835690.

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

# A Appendix

## A.1 Fused Gromov-Wasserstein (FGW)

The algorithm to calculate the fused Gromov-Wasserstein distance between two views in our decoder is provided as pseudo-code in Algorithm 1. The algorithm takes $\boldsymbol{C}^{(v)}$, $\boldsymbol{C}^{(v')}$, and $\boldsymbol{C}^{(vv')}$, which are intra/inter-costs between nodes in a matrix form, as well as $\rho$, which is a hyper-parameter. It returns the Wasserstein distance, Gromov-Wasserstein distance, as well as the transport matrix.

---

**Algorithm 1:** Computing fused Gromov-Wasserstein distance.

---

1  **Input:** $\boldsymbol{C}^{(v)}_{n\times n}$, $\boldsymbol{C}^{(v')}_{m\times m}$, $\boldsymbol{C}^{(vv')}_{n\times m}$, $\rho$
2  **Definitions:** $\odot$ = Hadamard product, $\langle\cdot,\cdot\rangle$ = Frobenius dot-product

3  // Cross-view similarity:
4  $\quad \hat{\boldsymbol{C}}^{(vv')} = (\boldsymbol{C}^{(v)})^2 \mathbf{1_n}\mathbf{1_m}^\top + \mathbf{1_n}\mathbf{1_m}^\top((\boldsymbol{C}^{(v')})^2)^\top$
5  // Initializing variables:
6  $\quad \boldsymbol{T} = \mathbf{1_n}\mathbf{1_m}^\top, \quad \boldsymbol{\sigma} = \frac{1}{m}\mathbf{1_m}, \quad \boldsymbol{B}_{i,j} = \exp(\hat{\boldsymbol{C}}^{(vv')}_{i,j})/\rho$
7  **for** $t_1 = 1, 2, \ldots$ **do**
8  $\quad\quad \mathcal{L} = \hat{\boldsymbol{C}}^{(vv')} - 2\boldsymbol{C}^{(v)}\boldsymbol{T}(\boldsymbol{C}^{(v')})^\top$
9  $\quad\quad$ **for** $t_2 = 1, 2, \ldots$ **do**
10 $\quad\quad\quad \boldsymbol{M} = \boldsymbol{B} \odot \boldsymbol{T}$
11 $\quad\quad\quad$ **for** $t_3 = 1, 2, \ldots$ **do**
12 $\quad\quad\quad\quad \boldsymbol{\delta} = \frac{1}{n\boldsymbol{M}\boldsymbol{\sigma}}, \quad \boldsymbol{\sigma} = \frac{1}{n\boldsymbol{M}^\top\boldsymbol{\delta}}$
13 $\quad\quad\quad \boldsymbol{T} = \operatorname{diag}(\boldsymbol{\delta})\,\boldsymbol{M}\,\operatorname{diag}(\boldsymbol{\sigma})$

14 $\mathcal{D}_W = \langle(\boldsymbol{C}^{(vv')})^\top, \boldsymbol{T}\rangle$
15 $\mathcal{D}_{GW} = \langle\mathcal{L}^\top, \boldsymbol{T}\rangle$

16 **Return** $\boldsymbol{T}, \mathcal{D}_W, \mathcal{D}_{GW}$

---

## A.2 Data description

**Microbiome-metabolome interactions.** The goal studying this dataset is to detect the microbe-metabolite interactions in patients with Cystic Fibrosis (CF). This dataset includes the 16S ribosomal RNA (rRNA) sequencing and metabolomics for 172 patients diagnosed with CF. We follow the same preprocessing steps as in Morton et al. (2019); Hajiramezanali et al. (2020), and filter out microbes that appear in less than ten samples, which results in 138 unique microbial taxa and 462 metabolite features. To construct the microbiome network, we perform a taxonomic enrichment analysis using Fisher's test and calculating p-values for each pairs of microbes as in Hajiramezanali et al. (2020). More specifically, the Benjamini-Hochberg procedure (Benjamini & Hochberg, 1995) is adopted for multiple test correction and an edge is added between two microbes if the adjusted p-value is lower than 0.01, The microbiome graph has 984 edges with the graph density of 0.102. For the metabolomics network, there are 1185 edges in total, with each edge representing a connection between metabolites via a same chemical construction (Morton et al., 2019). The graph density of the metabolite network is 0.011. We use $80\%$ of the reported target molecules of *P. aeruginosain* studies in Quinn et al. (2015) and Morton et al. (2019) as a test set to evaluate the predicted microbiome-metabolome interactions. The remaining $20\%$ of the reported molecules are considered as a validation set and are only used for the early stopping purpose.

**Precision medicine.** Here we aim to identify genetic markers of cancer drug responses. This is a very challenging task due to the very limited number of observations with respect to the system complexity and huge number of biological and experimental confounders, which often leads to significant false positive associations (Barretina et al., 2012). We consider a dataset from 30 acute myeloid leukemia (AML) patients that contains gene expression and drug sensitivity data of 160 chemotherapy drugs and targeted inhibitors (Lee et al., 2018). For gene expression, we preprocessed the RNA-Seq data resulting in 9073 genes (Lee et al., 2018). Following Hajiramezanali et al. (2020), we construct the gene regulatory network based on the publicly available expression data of the 14 AML cell lines from

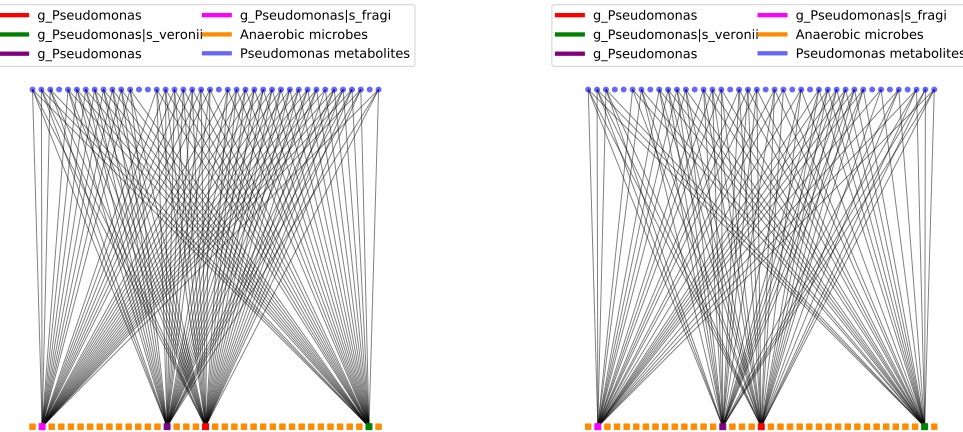

Figure 2: A sub-network of the relational graph consisting of *P. aeruginosa* microbes, their validated targets, and anaerobic microbes, inferred using MoReL_{ss} (**Left**) and MoReL_{us} (**Right**) with sub-network negative accuracy of 100%.

the Cancer Cell Line Encyclopedia1 (CCLE) using R package GENIE3 (Vân Anh Huynh-Thu et al., 2010). Moreover, We construct drug-drug interaction networks based on their action mechanisms. Specifically, the selected 53 drugs are categorized into 20 broad pharmacodynamics classes (Lee et al., 2018); 14 classes contain more than one drugs. Only 16 out of the 53 drugs are shared across two classes. We consider that two drugs interact if they belong to the same class. We use the area under the drug response curve reported in the CCLE dataset to indicate drug sensitivity across a range of drug concentrations (Barretina et al., 2012; Lee et al., 2018). Following Lee et al. (2018), we only consider the drugs that have less than 50% cell viability in at least half of the samples, resulting in 53 drugs. We use 797 reported drug-gene interactions in The Drug–Gene Interaction Database (DGIdb) (Wagner et al., 2016) in order to evaluate different models. We note that our test and validation sets only include the interactions for 43 of the 53 drugs in the dataset. We use 20% of the evaluation set as the validation set. Please note that the validation set has been only used for early stopping.

### A.3    Additional results for CF dataset

In this section, we provide additional results for CF dataset demonstrating interpretability and robustness of MoReL.

Figure 2 shows two sub-networks of the inferred bipartite relational graphs by MoReL_{ss} and MoReL_{us}, consisting *P. aeruginosa*, anaerobic microbes, and validated target nodes of *P. aeruginosa* and all of the inferred interactions between them. Based on the biology knowledge, the expected interactions in these sub-networks should be that the four highlighted nodes in the bottom row are connected to all of the nodes in the top row, and any other nodes in the bottom row are not connected to any of the top nodes. At the sub-network negative accuracy of 100% (i.e. any nodes in the bottom row other than the four highlighted ones are not connected to any of the top nodes), while MoReL_{us} identifies 70% of the validated edges of *P. aeruginosa*, MoReL_{ss} identifies 86.8% of the edges. We note that BayReL identifies 78% of the validated interactions (Hajiramezanali et al., 2020). This clearly shows the effectiveness of our proposed FGW-based decoder and interpretability of MoReL to identify inter-relations.

In the main manuscript, we only reported the results for one specific threshold value of negative accuracy (97%). Here we provide additional results with other threshold values, which show similar improvements over competing methods and similar trends by MoReL_{ss} and MoReL_{us}, as clearly observed in Figure 3. We note that there is a trade-off between positive and negative accuracy, and the optimal point can be chosen depending on the application.

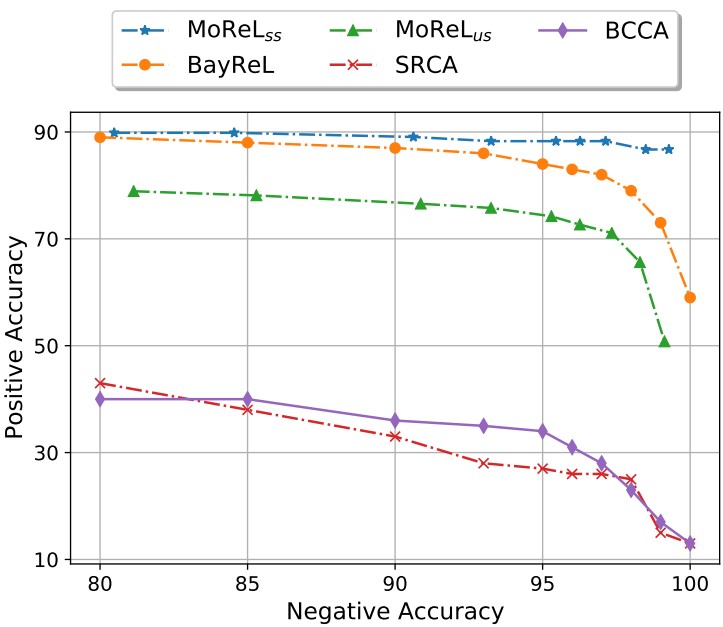

Figure 3: Positive accuracy vs negative accuracy of various models in CF data.

