# OpenReview forum: "MoReL: Multi-omics Relational Learning"
_ICLR.cc/2022/Conference — ICLR 2022 Poster_

### Official Review · Reviewer_wNNw · 2021-11-03

**Correctness:** 3
**Technical Novelty And Significance:** 3
**Empirical Novelty And Significance:** 3
**Recommendation:** 8
**Confidence:** 3

**Main Review:**

On the positive side: the approach deals with problems that could not be previously easily tackled, such as 1) having access to a graph of dependencies between features for each view; 2) dealing with missing instances on some of the views; 3) having pairing information across each view.
On the negative side: a more thorough empirical investigation of how well the approach can deal with those cases is lacking. Specifically, 1) a study of the robustness when the graph of dependencies between features is noisy is missing (one should expect that some edges are wrong and study the dependency of the performance w.r.t. the number of wrong edges); 2) the authors show a single example where 10% of data is removed, however a study that reports the performance w.r.t. the fraction of removed data is needed to substantiate the claim that the approach is indeed successful in dealing with missing data; moreover the fraction of missing instances should be different for different views; 3) the authors report a single example of lack of alignment (reversing the order for one view but not the other), while a more systematic analysis with several random permutations is what should be reported.
The computational complexity is reported for a single dataset, while it should be possible to give a sense of the time dependency of the proposed approach  w.r.t. the training set and test set size (does it scale linearly?).
Some comments would be appreciated to clarify how the model deals with the case of several structured or unstructured views: arguably the  graph of dependencies between features would form disconnected components when multiple structured views are considered and hence the shortest path distance cannot be computed in that case.
With regards to the case with multiple views, it would be of interest to study the sample complexity as a function of the number of available views, since, as the authors have stated in the introduction, leveraging the information from multiple sources should improve the performance on tasks with a limited number of training samples.


**Summary Of The Paper:**

The authors propose a Bayesian model to infer relations across heterogeneous views of data, which can be of structured and unstructured type.

**Summary Of The Review:**

The approach deals with problems that could not be previously easily tackled, however  a more thorough empirical investigation of how well the approach can deal with those cases is lacking.

---

> ### Author Response · Authors · 2021-11-17
> **Response to reviewer wNNw**
>
> We thank the reviewer for the time and efforts reviewing our paper.
>
> **Ablation studies**:
> We conducted the ablation studies suggested by the reviewer on the CF dataset. We assumed that the microbiome view is structured while the metabolite view is unstructured.
>
> 1- ***Edge perturbation***: We randomly removed edges in microbiome view while assuming that  metabolite view is unstructured. The positive accuracies at negative accuracy of 97% are reported in the following table:
>
> | % of removed edges | 0     | 10    | 25    | 50    |
> |--------------------|-------|-------|-------|-------|
> | MoReL$_{us}$         | 63.77 | 61.70 | 60.41 | 58.63 |
> |                    |       |       |       |       |
> |                    |       |       |       |       |
>
> The performance consistently drops as we remove edges. This is expected as we are removing meaningful edges that may help the model derive better node embeddings.
>
> 2- ***Missing samples***: We randomly removed samples in one view while keeping all the samples from the other view. The positive accuracies at negative accuracy of 97% are reported in the following table:
>
>
> |     | metabolite | microbiome |
> |-----|------------|------------|
> | 0%  | 63.77      | 63.77      |
> | 10% | 62.51      | 61.36      |
> | 30% | 59.68      | 58.75      |
> | 50% | 52.30      | 51.62      |
>
> The performance consistently drops as we remove samples. The drop is more significant from 30% to 50%. Reducing the very limited number of samples (172) by half, makes the faithful learning of the posterior distribution of embedding representations more challenging, hence decoder cannot achieve the same accuracy for relational learning when aligning learned distributions .
>
> 3- ***Sample permutation***: We randomly permuted the samples in microbiome view (5 different initialization) while keeping the order of samples in microbiome view. The positive accuracy (at negative accuracy of 97%) is 63.90 +/- 1.5. We can see that the accuracy did not change (compared to aligned samples) because of our distribution-based relational learning enabled by the FGW-based decoder.
>
> **Computational complexity**:
> Generally speaking, the FGW-based decoder is the bottleneck in terms of computational complexity. Therefore, the number of nodes/features in views has a more significant effect on training time compared to the number of samples. Each training epoch of MoReL for CF datasets takes 2.7 seconds while for precision medicine experiments it takes 11.9 seconds. Please note that the number of nodes/features in precision medicine is significantly larger than the CF dataset. A few methods could be used to further reduce the training time. For example, a hierarchical alignment scheme with graph pooling in each view would reduce the training time. In the current submission, we have demonstrated that MoReL leverages FGW-based decoder for distribution-based relational learning to reliably infer potential interactions across different types of molecules when we have heterogeneous data (graph-structured or not, without sample correspondence (unpaired samples) across views, potential missing or permutated samples, etc.), as shown in our presented experimental results. The scalability and time efficiency of MoReL will be further investigated and improved when we have the corresponding large-scale appropriately annotated data ready for the analyses.
>
> **Shortest path in disconnected graphs**:
> Most of the input graphs in our dataset are disconnected. Hence, the shortest path between some nodes is infinite. In our code, we replace “inf”s in the shortest path matrix with a large number (relative to the largest shortest path). In both datasets, we replaced “inf”s with 12. We will revise the section of implementation details accordingly.
>
> **Multi-view sample efficiency**:
> We would appreciate the references to the corresponding multi-view (more than two) omics data with appropriate annotations to test MoReL. We will further evaluate the scalability and computational efficiency of MoReL with the corresponding large-scale data considering the complexity of quantifying performance of relational learning models on datasets with more than two views.

---

### Official Review · Reviewer_cXWm · 2021-11-04

**Correctness:** 3
**Technical Novelty And Significance:** 3
**Empirical Novelty And Significance:** 3
**Recommendation:** 6
**Confidence:** 3

**Main Review:**

Strengths:

I generally find the paper quite well written. The paper may be interesting to a much broader community rather than just the biomedical use cases the experiments have shown, as the methodology is presented for any multi-omic data in a very general way.

The overall methodology is similar to BayRel. Both papers formulate the problem of multi-omics data integration by modeling it as multi-view link prediction. However, this paper does have several nontrivial improvements over BayRel: it could handle scenarios that have no known structure information, and the samples could be unpaired across views.

Weaknesses:
My main concern is with the experimental part.
The experiments are not extensive and cannot fully demonstrate the advantages of the proposed approach. First, the datasets are restrictive. Microbiome-metabolite interactions in CF may not be the most commonly used benchmark datasets in the biomedical multi-omic datasets, with many other more usual datasets not considered. Second, many of the advantages the proposed method claims cannot be fully demonstrated by the two datasets in experiment, though the authors do manipulate the structure of datasets manually to show that the method still works when some key assumptions of BayRel do not hold. It should be more telling if the authors could conduct a more comprehensive experimental studies, including more common datasets and datasets that are naturally with missingness and pairing issues. Also as the computational overhead compared to BayRel, knowing whether the methods could handle large-scale data with heterogeneous dependency structures would be helpful.


**Summary Of The Paper:**

The authors propose a Bayesian framework to learn relations among multi-omic datasets. The main unique advantage over existing methods is the proposed method is able to learn without a priori dependency structure and it allows a certain degree of missingness and mismatching. Experiments on two biomedical multi-omics data sets partially demonstrate the effectiveness of the proposed method.

**Summary Of The Review:**

The proposed method is convincing with respect to an important problem. The paper has nontrivial improvement over existing methods. Though it lacks comprehensive experimental validation, it is overall a good complement to existing literature.

---

> ### Author Response · Authors · 2021-11-17
> **Response to reviewer cXWm**
>
> We thank the reviewer for the time and efforts reviewing our paper.
>
> **Datasets**:
> To be able to compare our MoReL with BayReL, we used datasets that are naturally structured in all views. Therefore, we used the same datasets introduced in the BayReL paper. We would truly appreciate the references from the reviewer to the suggested datasets with mixed types of views so that we can test MoReL on them.
>
> **Computational complexity**:
> Generally speaking, the FGW-based decoder is the bottleneck in terms of computational complexity. Therefore, the number of nodes/features in views has a more significant effect on training time compared to the number of samples. Each training epoch of MoReL for CF datasets takes 2.7 seconds while for precision medicine experiments it takes 11.9 seconds. Please note that the number of nodes/features in precision medicine is significantly larger than the CF dataset. A few methods could be used to further reduce the training time. For example, a hierarchical alignment scheme with graph pooling in each view would reduce the training time. In the current submission, we have demonstrated that MoReL leverages FGW-based decoder for distribution-based relational learning to reliably infer potential interactions across different types of molecules when we have heterogeneous data (graph-structured or not, without sample correspondence (unpaired samples) across views, potential missing or permutated samples, etc.), as shown in our presented experimental results. The scalability and time efficiency of MoReL will be further investigated and improved when we have the corresponding large-scale appropriately annotated data ready for the analyses.

---

### Official Review · Reviewer_dcdZ · 2021-11-04

**Correctness:** 3
**Technical Novelty And Significance:** 2
**Empirical Novelty And Significance:** 2
**Recommendation:** 5
**Confidence:** 2

**Main Review:**

The paper is dense and terse making it difficult to follow in order to pick out the novelty of the proposed approach. The definition of the experiments and the data used in the Experiments section is not self-sufficient. Without consulting the Appendix it is hard to know what are the problems being solved, the preprocessing and representation of the data to be used as input to the method. Although the approach is aimed at multiomic integration, there is no real connection to the domain of application. The evaluation is focused on the comparison to BayReL a related approach mentioned throughout the paper but not described in the related work.

Multi-omics integration is of interest in computational biomedicine. However, focusing on BayReL only, in the related work the authors miss a large body of work, old and new. To name some: SNF, iCluster, LIGER, embedding propagation. They also miss out on a large amount of data being generated with increasing pace within the field.

Judging from the results, MoReL seems to offer some performance improvement over BayReL on structured problems. The authors demonstrate the ability of MoReL to learn from unpaired data and in a setting with missing data. It all comes at a cost of 4.5 fold increase in computational time.


**Summary Of The Paper:**

The authors propose a hierarchical Bayesian generative model for multiview learning aimed at application to integration of multiple modalities of omics data.


**Summary Of The Review:**

While there is a limited novelty and demonstrated improvement in performance over a single related approach, from the application and result interpretation point of view, the advantages of using MoReL are not clear.

---

> ### Author Response · Authors · 2021-11-17
> **Response to reviewer dcdZ**
>
> We thank the reviewer for the time and efforts reviewing our paper.
>
> **BayReL and data description**:
> We tried to describe BayReL (in introduction and in Section 5.4) and data description concisely in the main manuscript. We will make sure to have a detailed description in the main manuscript considering available space.
>
> **Connection to application**:
> Our main focus in MoReL is developing a computational method for finding relations between features across heterogenous views. Indeed, MoReL is a general method that can potentially be applied to other applications, but the problem setting that we are addressing mostly appears in biological datasets and specially in multi-omics data integration.
>
> **Related works**:
> The reason that there is an emphasis on comparison with BayReL is that to the best of our knowledge, BayReL is the only graph neural network based data integration method that learns “relations” between features across (graph-structured) views. Other methods address different  problems that can appear similar but not exactly the same. For example, SNF and LIGER learn the relations between samples/patients not features, while iCluster is a multi-view clustering algorithm. The vast majority of data integration methods are usually interested in finding a shared latent space which needs ad-hoc post processing to find relations. We thank the reviewer for pointing out these works in the literature and we will make sure to have a more comprehensive section of related works considering the available space.
>
>
> **Novelty and improvements**:
> MoReL is the first relational learning method that can integrate both structured and unstructured heterogeneous views in one framework. More specifically, MoReL learns the relations between features across different views of data by exploiting non-linear transformations of data while capturing view-specific structure information if available. MoReL solves a difficult problem without any restrictive assumptions which makes it applicable to more realistic real-world scenarios. Specifically in our problem setting: 1) Each view can be structured or unstructured, 2) There can be missing samples in views, 3) Samples can be misaligned. These are very common issues in multi-omics datasets especially when feature measurements across views are collected from different subjects/samples by different labs. We want to emphasize that BayReL cannot address the problem when any of the aforementioned assumptions hold which makes it unsuitable for real-world applications. The FGW-based decoder in MoReL allows finding relations through distribution-based alignment (instead of distance-based alignment in the embedding space) and contributes to significant performance improvement compared to BayReL as discussed in details in Section 5.4.

---

> > ### Comment · Reviewer_dcdZ · 2021-11-24
> > **Response to authors**
> >
> > Data and application: True, the problem appears in biology in the context of multi-omics integration. The comment about the problem being hard to understand, relates to them not being described properly in the main text and not having a clearly established connection with the proposed aoproach. In this way the aplication, although a central part of the paper, seems not to be as relevant. The claim about the generality of the approach, given the current work, can't be made, as there is no proof or experimental evaluation to support it.
> >
> > Related work: Extending the related work and placing the proposed approach in this context will improve the manuscript and will make the impact of the work clearer to establish.
> >
> > Novelty and improvements: I partially agree with the comments. The applicability to real world scenarios, especially to more recent, more complex and richer single cell data, might not be possible due to the reported computational complexity as noted in my original review.
> >
> > My evaluation score remains the same.

---

> > > ### Author Response · Authors · 2021-11-24
> > > **Re: Response to authors**
> > >
> > > We appreciate the reviewer’s feedback. Regarding the suggestions to extend related works and better present the context of the problem settings with connection to existing methods, we tried our best in our submission and in our responses to clearly describe them. We truly appreciate it if the reviewer can point out which aspects of the problem settings are not "*described properly*" so that we can further improve the presentation.
> > >
> > > Regarding the generalizability to more complex data, we truly believe that MoReL is capable of doing what we stated, based on the presented experiments in the original submission as well as the additional ablation studies based on the suggestions of Reviewer wNNw (that can be found in our response to Reviewer wNNw). We again would truly appreciate the help from the reviewer to point us to related benchmark datasets so that we can test MoReL. We also agree that the current implementation can be further improved to make MoReL more computationally efficient and scalable, including the potential hierarchical alignment procedure as we mentioned in our response to Reviewers cXWm and wNNw.

---

### Official Review · Reviewer_2ULW · 2021-11-08

**Correctness:** 3
**Technical Novelty And Significance:** 2
**Empirical Novelty And Significance:** 3
**Recommendation:** 5
**Confidence:** 3

**Main Review:**

The proposed model allows to integrate of both structured and unstructured heterogeneous views which often happens in the context of omics data. In order to align nodes/features in every pair of views, the Gromov-Wasserstein (FGW) has been used. The experimental results show a significant improvement of the proposed model to the baselines in several downstream tasks. However, the paper can be improved in the following aspects:
  - It is not clear what is the key difference between BayReL and the proposed besides the FGW regularization term.
  - Can we apply BayReL and FGW into a pipeline to learn heterogeneous views? And how is it compared to the proposed models if possible?
  - What is the square node FGW in the graphical model in Figure 1. It is difficult to follow the generative process of the proposed models.

**Summary Of The Paper:**

The paper proposed a deep Bayesian model for heterogeneous multi-omics data integration. The Gromov-Wasserstein (FGW) regularization between latent representations of heterogeneous views is used to align nodes/features in every pair of views. The experimental results have demonstrated improvement in inferring meaningful relations.

**Summary Of The Review:**

 In short, the paper is trying to solve an important challenge of heterogeneous multi-omics data integration by proposing an advanced model compared with BayReL. However, what is the key difference and improvements compared with BayReL should be discussed in detail.

---

> ### Author Response · Authors · 2021-11-17
> **Response to reviewer 2ULW**
>
> We thank the reviewer for the time and efforts reviewing our paper.
>
> **Differences from BayReL**:
> As pointed out in Introduction and Section 5.4 of submission:
>
> 1- The problem formulation in MoReL is different from BayReL. MoReL solves the relational learning problem with more realistic and less restricting assumptions compared to BayReL. More specifically, BayReL assumes that: 1) All views are graph structured, 2) There are no missing samples in any of the views, 3) Samples are completely aligned. These indeed are not realistic/practical assumptions, making BayReL sometimes not directly applicable to several real-world scenarios as explained in the submission.
>
> 2- We want to emphasize that in MoReL, we use a “FGW-based decoder” as discussed in Section 4.2.1 of the submission (we are NOT just adding a FGW regularization term to the loss function). In fact, The FGW block in Figure 1 refers to “FGW-based decoder”, which enables *distribution-based* relational learning. We will revise the manuscript accordingly to make it clearer. In BayReL, the same encoder is used for all views and the embeddings are matched using a simple distance-based decoder. Adding an FGW regularization term does not solve the aforementioned issues because: 1) the same encoder cannot be applied to structured and unstructured views or when two views have different numbers of samples, 2) the decoder cannot handle the arbitrary permutation/rotation of latent dimension (caused by misalignment of input) because the decoder is still *distance-based*.
>
> 3- Our detailed ablation study (Section 5.4) in the submission showed the superior performance of MoReL in multiple problem settings (including the BayReL problem setting) and higher expressive power of MoReL with distribution-based alignment in our FGW-based decoder. We want to re-emphasize that the key distinction of MoReL and BayReL is that MoReL solves a more difficult and realistic problem. The aforementioned issues are common in multi-omics datasets especially when feature measurements across views are collected from different subjects/samples by different labs.

---

### Decision · Program_Chairs · 2022-01-20

**Decision:**

Accept (Poster)

**Comment:**

A deep Bayesian generative model is presented for multi-omics
integration, using fused Gromov-Wasserstein regularization between
latent representations of the data views. The method removes several
non-trivial and practically important restrictions from an earlier
method BayRel, enabling application in new setups, while still
performing well.

Reviewers discussed the paper with the authors, resolving
misunderstandings of the differences from earlier work
(esp. BayReL). The authors reported more extensive experiments in the
rebuttal, though not comparisons. The main remaining weakness is that
the contributions are in a very narrow field, or at least aplications
have only been demonstrated in the narrow field of multi-omics data
analysis. And even within that field, only in a narrow subfield. In a
machine learning venue that is restrictive. Another issue is
computational efficiency. The final decision then depends on how much
weight we place on the novel contributions vs these weaknesses.